# Revealing hidden spin polarization in centrosymmetric van der Waals materials on ultrafast timescales

B. Arnoldi[1], S. L. Zachritz[2], S. Hedwig[1], M. Aeschlimann [1], O. L. A. Monti[2,3] ✉ & B. Stadtmüller [1,4] ✉

One of the key challenges for spintronic and quantum technologies is to achieve active control of the spin angular momentum of electrons in nanoscale materials on ultrafast, femtosecond timescales. While conventional ferromagnetic materials and materials supporting spin texture suffer both from conceptional limitations in miniaturization and inefficiency of optical and electronic manipulation, non-magnetic centrosymmetric layered materials with hidden spin polarization may offer an alternative pathway to manipulate the spin degree of freedom by external stimuli. Here we demonstrate an approach for generating transient spin polarization on a femtosecond time-scale in the otherwise spin-unpolarized band structure of the centrosymmetric 2H-stacked group VI transition metal dichalcogenide $WSe_2$. Using ultrafast optical excitation of a fullerene layer grown on top of $WSe_2$, we trigger an ultrafast interlayer electron transfer from the fullerene layer into the $WSe_2$ crystal. The resulting transient charging of the $C_{60}/WSe_2$ interface leads to a substantial interfacial electric field that by means of spin-layer-valley locking ultimately creates ultrafast spin polarization without the need of an external magnetic field. Our findings open a novel pathway for true optical engineering of spin functionalities such as the sub-picosecond generation and manipulation of ultrafast spin currents in 2D heterostructures.

Fundamental to the advance of spintronics and the creation of novel quantum functionalities in solids is the ability to encode, manipulate, and store information onto the spin angular momentum of electrons with high efficiency and low volatility[1,2]. Ferromagnetic materials have long been the natural driving target for these efforts. However, their intrinsic limitations regarding miniaturization (i.e., governed by the super-paramagnetic limit) and their susceptibility to external stray fields have triggered a search for non-magnetic materials that can nevertheless support advanced spin functionalities.

One highly promising alternative to ferromagnets is non-magnetic bulk materials with broken symmetries and strong spin-orbit coupling. The combination of both properties results in the spin-splitting of bands in momentum space either through the Dresselhaus effect[3] or the bulk Rashba effect[4-6] and leads to intriguing spin functionalities such as the interconversion of charge and spin[7-9]. Unfortunately, while the resulting spin-momentum locking and associated spin texture do indeed enable controlling the spin degree of freedom, it also limits the type of spin operations that can be realized in such materials: For instance, an unpolarized charge current can only be converted into a

[1]Department of Physics and Research Center OPTIMAS, Rheinland-Pfälzische Technische Universität Kaiserslautern-Landau, Erwin-Schroedinger-Strasse 46, Kaiserslautern 67663, Germany. [2]Department of Chemistry and Biochemistry, University of Arizona, Tucson, AZ 85721, USA. [3]Department of Physics, University of Arizona, Tucson, AZ 85721, USA. [4]Institute of Physics, Johannes Gutenberg University Mainz, Staudingerweg 7, 55128 Mainz, Germany. ✉e-mail: monti@arizona.edu; b.stadtmueller@rptu.de

**Fig. 1 | Electronic valence band structure of the $C_{60}$/WSe$_2$ heterostructure.**
**a** Sketch of the local layer- and spin-dependent band structure of the two non-interacting WSe$_2$ layers of the bulk unit cell in which the spin polarization vanishes at every point in the Brillouin zone. **b** Illustration of the optical manipulation scheme for uncovering the hidden spin polarization of WSe$_2$. An ultrashort 3.2 eV laser pulse resonantly excites the ultrathin $C_{60}$ layer grown on top of WSe$_2$ leading to an ultrafast electron transfer into the first WSe$_2$ layer and to a transient E-field across the $C_{60}$/WSe$_2$ interface. **c** Energy vs. momentum photoemission map of the $C_{60}$/WSe$_2$ heterostructure along the Σ-K-direction (He I$_\alpha$ radiation). It shows the spin-split WSe$_2$ valence bands with their hole-like dispersion (VB$_1$, VB$_2$) and the

dispersion-less HOMO (H) of $C_{60}$. The right side of (**c**) shows the spin-resolved photoemission yield (out-of-plane spin component) of the valence band structure obtained at a selected electron momentum (see white dashed line). The red and blue curves represent the fit to the spin-up and spin-down spectrum, respectively. The contributions of the first and second-layer valence bands to the spectral yield are fitted and illustrated as green and blue Gaussian curves underneath the spectra. The different photoemission intensity of the valence band of the first and second WSe$_2$ layer is due to the small elastic mean free path of the photoelectrons at small kinetic energies leading to an exponential attenuation of the photoemission signal with increasing distance from the surface.

transverse spin current by the spin Hall effect[10]. In addition, electric and optical gating, needed for fast operations, can only manipulate the magnitude of the momentum-dependent spin splitting, and both are limited to the picosecond timescale due to the intrinsic buildup time of the photovoltage[11,12]. These fundamental limitations underline the need for new paradigms to tailor and manipulate the spin degree of freedom, ideally by directly creating and manipulating spin polarization rather than spin texture.

In this regard, the discovery of the so-called hidden spin polarization in non-magnetic materials with centrosymmetric crystal symmetry suggests a pathway toward realizing spin manipulation in a much larger class of materials[13–16]. Hidden spin polarizations emerge in centrosymmetric layered structures containing subunits with broken inversion symmetry. Typical examples are, for instance, 2H-stacked group VI transition metal dichalcogenides (TMDs), of which one of the most prominent examples 2H-WSe$_2$ is the focus of the present study. A cartoon of the salient features of the spin- and layer-dependent valence band structure of this material is shown in Fig. 1a: It is characterized by spin-split valence bands, localized within each individual layer of the 2H-stacked structure[14,17–19], and whose spin is reversed between the valleys at the high symmetry point K and its time-reversal couple K'. Inversion symmetry of the full bulk unit cell, which contains two layers, leads to an inversion of the valence band spin polarization at each high symmetry point in successive layers, resulting as expected in an overall spin-degenerate bulk band structure. If however the inversion symmetry in otherwise centrosymmetric 2H-WSe$_2$ can be broken between two adjacent layers, e.g., by addressing individual layers differentially, then the emergence of previously hidden spin polarization may be expected, enabling manipulation of spin degree of freedom without magnetic fields and potentially on ultrafast timescales.

In this work and by using spin- and time-resolved angle-resolved photoemission spectroscopy (ARPES), we overcome this challenge and demonstrate a new approach to generate transient spin polarization by lifting the spin degeneracy of the bulk band structure at the interface of a $C_{60}$/2H-WSe$_2$ heterostructure. Using ultrafast optical excitation, we are able to generate large interfacial electric fields that ultimately result in ultrafast spin polarization. Conceptionally, the influence of

such an interfacial field on the WSe$_2$ valence band structure can be illustrated by considering the Hamiltonian of the bare 2H-WSe$_2$ bulk crystal, even though it does not fully describe the electronic structure of the $C_{60}$/2H-WSe$_2$ heterostructure. It shows the coupled spin, spin-like valley, and layer pseudospin degrees of freedom that characterize the hidden spin polarization of 2H-WSe$_2$ in the valence band and near the K-points[17]:

$$H_v = -\lambda_v \tau_z s_z \sigma_z^v + t_\perp \sigma_x^v \qquad (1)$$

Here, the first term describes the coupling between the spin ($s_z$), valley-pseudospin ($\tau_z$), and layer-pseudospin ($\sigma_z^v$) degrees of freedom mediated by spin-orbit coupling $\lambda_v$ (SOC), and the second term describes the coupling of the weak interlayer hopping ($t_\perp$) in WSe$_2$ to the layer pseudospin. Importantly, carrier population in a specific layer represents an interlayer electronic polarization, and hence the layer pseudospin can be considered as an electrical polarizability that can mediate interactions between this spin-like quantity and an external, transient electric field via the Hamiltonian (1)[18–20].

In order to generate the layer-dependent ultrafast electric field, we take advantage of the unique properties of our hybrid organic/inorganic heterostructure by driving interfacial charge-transfer from $C_{60}$ to WSe$_2$ (Fig. 1b). The fullerene $C_{60}$ is ideally suited for this endeavor as its excited states spectrum is dominated by a manifold of so-called charge transfer excitons[21–23] that can act as precursor for charge separation and charge transfer processes across interfaces. The resulting transient band structure engineering by interfacial electric fields presents the first key step towards ultrafast generation of hole-like spin currents from an unpolarized DC-charge current running at the interface of TMD bulk materials by fs light excitation, without the need for large external magnetic fields, time-reversal or structural inversion symmetry breaking. Our conclusions are enabled by multi-dimensional photoemission spectroscopy of a $C_{60}$/WSe$_2$ hetero-structure with an ultrathin $C_{60}$ layer that allows us to directly access and uncover transient changes of the hidden spin polarization after optical excitation. In this way, we demonstrate that we are able for the first time to trace both the excited state and spin-dependent band structure dynamics at this hybrid heterointerface on the fs timescale.

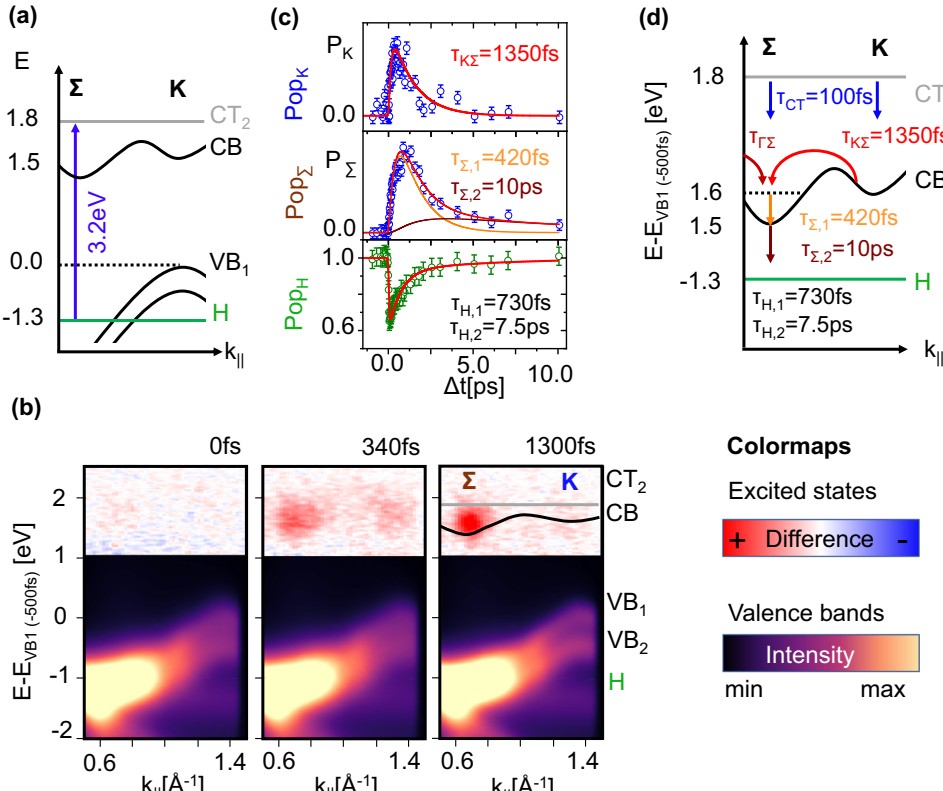

**Fig. 2 | Ultrafast electron and hole dynamics. a** Energy level diagram of the electronic band structures of the $C_{60}$/WSe$_2$ heterostructure. The blue arrow indicates the dominant optical transition of the 3.2 eV excitation. **b** Energy vs. momentum photoemission maps at selected pump-probe delays obtained with linearly polarized pump pulses (applied fluence $F = 0.5$ mJ/cm²). This rather large fluence is necessary to create the required charge density at the interface for the transient band structure engineering approach. The excited state region ($E - E_{VB} > 0$ eV) is shown as a difference map, and the valence band region as an electron intensity map (see colormaps). The energy and momentum positions of

the molecular CT$_2$ state and the WSe$_2$ valence band are superimposed onto the experimental data (1300 fs) as gray and black curves. **c** Temporal evolution of the WSe$_2$ excited state and $C_{60}$ HOMO intensity evolution. The error bars estimate the uncertainty of the fitting procedure to the experimental data. The solid lines superimposed onto the population dynamics at the K- and Σ-point (Pop$_K$ and Pop$_Σ$) were obtained by a rate equation model. The key scattering processes of this model are illustrated in (**d**), together with the scattering times of the best fit to the data. The temporal evolution of the HOMO is modeled with a double exponential fit function.

## Results

### Interfacial valence band structure

Our sample consists of an in situ prepared surface of a 2H-WSe$_2$ bulk crystal covered with ~0.8 ML of $C_{60}$ (see method section for more details). The energy level alignment of the valence band structure of the $C_{60}$/WSe$_2$ heterostructure prior to ultrafast excitation can be deduced from the momentum-resolved photoemission map in Fig. 1c, recorded along the Γ-Σ-K high symmetry direction and shown in the vicinity of the K-point. The spin-split valence bands of WSe$_2$ appear as hole-like parabolic features at the K-point with an energy splitting of 450 meV, similar to the bare WSe$_2$ surface (see Supplementary Figs. S1, S2 and ref. 14). Most importantly, we do not observe any modification of the band dispersion and no signature of a possible Rashba-like spin splitting of the WSe$_2$ valence band structure (neither at the Γ- or K-point, see Supplementary Fig. S1) due to the broken inversion symmetry at the $C_{60}$/WSe$_2$ interface. All this is indicative of physisorptive interactions at the $C_{60}$/WSe$_2$ interface. The non-dispersive feature at $E - E_{VB} = -1.3$ eV is attributed to the $C_{60}$ valence state, i.e., the highest occupied molecular orbital (HOMO), and reflects the large ionization energy of $C_{60}$[24,25]. The spin-resolved photoemission yield of the valence band structure is shown on the right for a selected electron momentum (indicated by a white vertical line in Fig. 1c). The red curve corresponds to the yield of spin-up electrons (out-of-plane spin direction), and the blue curve to the yield of spin-down electrons. The $C_{60}$ HOMO (H) is not spin-polarized, as expected for molecular films on non-magnetic surfaces. In contrast, we find

strong out-of-plane spin polarization for both SOC-split WSe$_2$ valence bands (VB$_1$ and VB$_2$). Though bulk WSe$_2$ does not support a spin-split density of states[13], the layer-dependent hidden spin polarization of inversion-symmetric bulk WSe$_2$ is made apparent by the extreme surface sensitivity of the photoemission process[14]. In ARPES, we primarily probe the top WSe$_2$ layer due to the small elastic mean free path of ~1 nm for WSe$_2$ leading to an exponential attenuation of the photoemission signal with increasing distance to the surface[14,26]. Hence the photoemission yield at VB$_1$ carries mostly spin-up electrons (green curve/area) near K, while the smaller signal in the spin-down channel (blue curve/area) stems from the inverted spin-polarization of VB$_1$ of the second layer. The opposite is true for the lower valence band VB$_2$. Key to these observations is the fact that our spin- and momentum-resolved photoemission experiment also provides layer sensitivity. This allows us to disentangle ultrafast momentum-, spin- and layer-dependent band structure changes in the $C_{60}$/WSe$_2$ heterostructure following optical excitation.

### Ultrafast interfacial charge transfer dynamics

Optical excitation of the heterostructure with 3.2 eV sub-50 fs pulses at a sufficient applied fluence of ~0.5 mJ/cm² creates a transient electric field across the $C_{60}$/WSe$_2$ interface: At this photon energy, the CT$_2$ state of $C_{60}$ is excited (see energy level alignment diagram in Fig. 2a), supported by previous studies of the two materials[20–23]. This state is associated with the formation of intermolecular charge transfer excitons. Using time- and momentum-resolved photoemission, we follow

the ultrafast charge-carrier dynamics subsequent to excitation at 3.2 eV. Example energy vs. momentum cuts from these data are shown in Fig. 2b at three characteristic time delays, showing clearly interfacial charge transfer from $C_{60}$ into the $WSe_2$ layer followed by scattering in the $WSe_2$ CB. The experimental data in the excited states are plotted as difference maps (accumulation of spectral yield shown in red, depletion in blue), while the transient changes in the valence band region are shown as intensity maps. Upon excitation ($t = 0$ fs), a broad distribution in momentum space is created at the energy of the $CT_2$ state, accompanied by an instantaneous intensity reduction of the HOMO feature. These optically induced modifications of the electron and hole population coincide with transient linewidth broadening of the interfacial valence band structure which was recently identified as a spectroscopic signature of charge-transfer excitons in molecular films[21,22]. Crucially, we only observe a marginal depletion of the $WSe_2$ valence states (see Supplementary Figs. S4 and S6). This proves that the formation of charge-transfer excitons in the $C_{60}$ layer is indeed the dominant optical excitation path and that direct excitation of $WSe_2$[27] does not play a dominant role here.

This broad electron distribution ($t = 0$ fs) evolves to populating the K- and $\Sigma$-valley of the $WSe_2$ conduction band ($t = 340$ fs), clearly indicating ultrafast electron transfer from $C_{60}$ to $WSe_2$ as previously observed for other molecule/TMD heterostructures[28–30]. Subsequently, phonon-mediated intervalley scattering redistributes carriers from the K-valley into the $\Sigma$-valley of $WSe_2$ ($t = 1300$ fs), a well-known process in TMDs[20,31,32].

To gain a more quantitative understanding of the interlayer and intervalley scattering processes at the $C_{60}/WSe_2$ interface, we model the changes in electron and hole populations using a rate-equation model and exponential fitting functions (see Supplementary Fig. S8 and Supplementary Methods A). We extract the transient electron population at the K- and $\Sigma$-valley of the $WSe_2$ conduction band and the hole population of the $C_{60}$ HOMO by analysis of the ARPES data (see Supplementary Figs. S4 and S5). The resulting traces for all three features are shown in Fig. 2c. Note that no clear population signal could be extracted for the broad $CT_2$ feature due to its large energetic overlap with the $WSe_2$ conduction band. The rate equation model considers the excited state scattering pathways illustrated in Fig. 2d. To model the interlayer charge transfer, we assume an initial population of the $CT_2$ state by the laser pulse, followed by electron-transfer processes from the $CT_2$ state into the K- and $\Sigma$-valley as well as intervalley scattering from the K- into the $\Sigma$-valley. The intrinsic electron dynamics of the $WSe_2$ bulk crystal are described by an initial laser-driven population of the $WSe_2$ conduction band at the $\Gamma$-point that scatters directly into the $\Sigma$-valley (see Supplementary Methods A).

Our model yields an interfacial electron transfer time of $\tau_{CT} = (100 \pm 50)$ fs (both for electron transfer into the K- and $\Sigma$-valley) with a substantially larger electron transfer from the molecular $CT_2$ state into the K-valley (see Supplementary Fig. S8). In addition, we find an intervalley scattering time of $\tau_{K\Sigma} = (1350 \pm 50)$ fs and the depopulation time of the electrons in the $\Sigma$-valley of about 10 ps. The intervalley scattering time $\tau_{K\Sigma}$ is ~20 times larger than previously reported for bare $WSe_2$[20]. We believe that this can be attributed to a sample temperature of ~40 K, much lower than in the previous report and causing significantly reduced electron-phonon scattering[33]. A detailed analysis of the hole population dynamics in Fig. 2c reveals a clear persistence of holes in the $C_{60}$ layer. Detailed analysis using exponential fit functions (discussed in ref. 21) shows instantaneous depletion of the $C_{60}$ HOMO within our experimental resolution, as expected from resonant excitation, followed by decay of the hole population in a two-step process with a fast recovery time constant of $(730 \pm 50)$ fs and a significantly slower second-time constant of ~7.5 ps.

Summarized in Fig. 3a, b, the key processes involve ultrafast interfacial charge transfer from $C_{60}$ to $WSe_2$, resulting in holes located on $C_{60}$ and electrons mainly in the K-valley of the top layer of $WSe_2$.

This is followed by intervalley scattering to $\Sigma$, whose electron density spans both the first and second layers of $WSe_2$[20]. This charge separation between $C_{60}$ and $WSe_2$ establishes a strong and transient interfacial electric field along the surface normal. As we show below, this field is ultimately responsible for revealing the hidden spin polarization in $WSe_2$. Eventually, the photoexcited electrons delocalize into the bulk $WSe_2$ crystal, and the interfacial electric field decays. Most importantly, the temporal evolution of this charge-separated state and the corresponding interfacial electric field is not influenced by the intrinsic electron dynamics of $WSe_2$ which predominantly leads to an electron population in the $\Sigma-$ valley exhibiting a delocalized electron density.

## Ultrafast changes in the interfacial energy level alignment

We next discuss the influence of the transient electric field on the interfacial energy level alignment. As can be seen in Fig. 3c, both $WSe_2$ valence band and the $C_{60}$ HOMO experience transient energy shifts, extracted from the energy distribution curves at K (see exemplary fits in Supplementary Fig. S5). We hence attribute these changes to transient Stark shifts caused by the large electric field built up by the interfacial charge transfer[34–38]. As expected for such Stark shifts, the sign of the shifts differs for the hole-enriched $C_{60}$ HOMO band and the valence band of the electron-enriched $WSe_2$: A simple electrostatic model of this interface[39] (see Supplementary Methods B and Supplementary Fig. S10 for details) that considers a distribution of holes residing on $C_{60}$ and electrons residing in the first $WSe_2$ layer is illustrated in Fig. 3d and reveals how the $WSe_2$ valence band and the $C_{60}$ HOMO features are expected to exhibit opposite energy shifts. It also shows that the magnitude and time evolution of the valence band shifts are determined by the transient number of charges in the adjacent interfacial layer. This behavior is fully confirmed by our experimental findings: The dynamics of the $WSe_2$ valence band shift follow the fast population dynamics of holes in the $C_{60}$ layer while its magnitude depends linearly on the number of holes within the $C_{60}$ layer (see Supplementary Fig. S9). Similarly, the valence band shift of the $C_{60}$ HOMO evolves on the timescale of the electron population in the K-valley where the electrons are localized in the first $WSe_2$ layer at the interface. Scattering of the electron from the K- into the $\Sigma$ valley is accompanied by delocalization into the 2nd layer, weakening the interfacial field. Both these observations and their explanation are also consistent with transient energetic shifts observed in recent experiments on the interface of $WS_2$ on graphene[37]. We conclude therefore that the observed dynamics are indeed driven by a layer-dependent interfacial Stark effect.

## Ultrafast generation of spin polarization in $WSe_2$

The interfacial electric fields hold the key to establishing layer-dependent transient spin polarization: The electric field experienced by the first and second $WSe_2$ layer differs and is coupled to the layer pseudospin, and since each layer is spin-valley-layer locked, the spin degeneracy of the bulk crystal is locally lifted in the first two layers. We investigate transient changes of the valence band spin polarization by monitoring the time evolution of the spin- and layer-dependent $WSe_2$ valence band structure in the vicinity of the K-point. The corresponding spin-dependent photoemission yield is shown in Fig. 4a for three characteristic instances in time, namely before the optical excitation ($t = -500$ fs), coincident with the optical excitation and initial formation of the charge transfer excitons in the $C_{60}$ layer ($t = 0$ fs), and in the presence of the charge-separated state at the $C_{60}/WSe_2$ interface ($t = 950$ fs). The spectra in the left column correspond to spin-up electrons, and those in the right column to spin-down. Fitting these spin- and time-resolved ARPES spectra reveals differential shifts of the valence bands $VB_1$ and $VB_2$ for the first (green Gaussian curve) and second (blue Gaussian curve) $WSe_2$ layer, summarized in Fig. 4b.

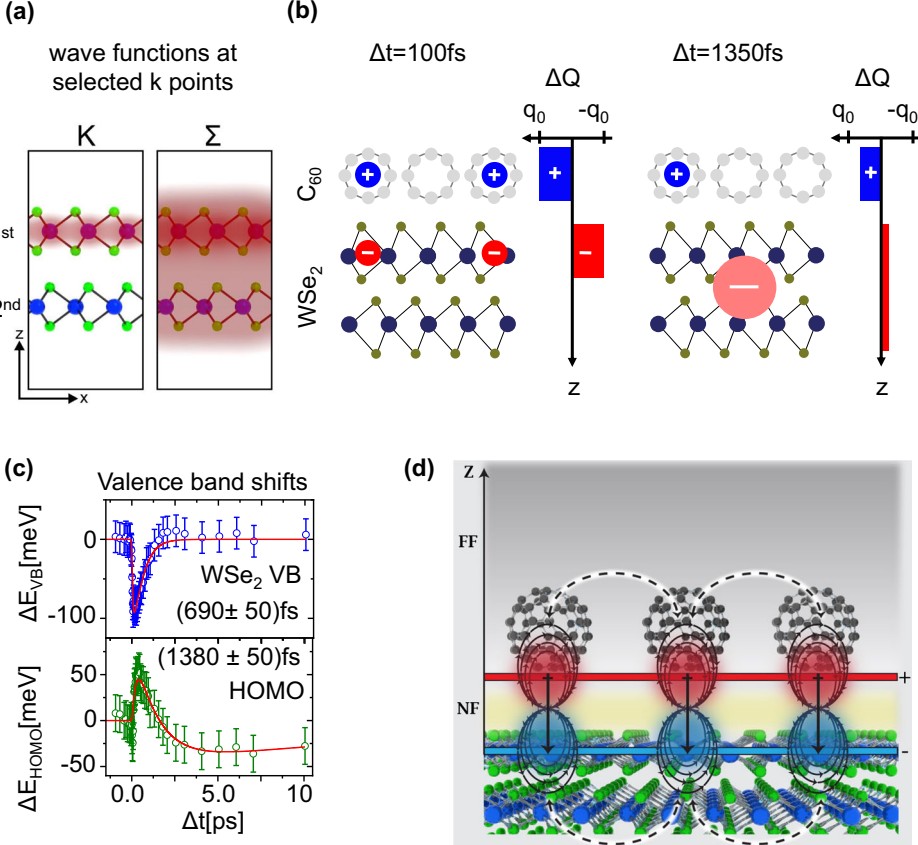

**Fig. 3 | Charge separation, interfacial E-field, and transient changes in the energy level alignment. a** Sketch of the real space electron densities (red shaded areas) of the wave functions at the K- and the Σ-valley of the WSe₂ conduction band (adapted from Bertoni et al. [20]). **b** Illustration of the charge separation process at the $C_{60}$/WSe₂ interface. After the ultrafast electron transfer from the $C_{60}$ CT₂ state into the WSe₂ K-valley, the electrons are confined to the first WSe₂ layer. Only the intervalley scattering form the K- into the Σ-valley leads to a delocalization of the electrons in WSe₂. **c** Temporal evolution of the valence band shifts of the WSe₂ (VB) and the $C_{60}$ (HOMO) valence states. The error bars estimate the uncertainty of the fitting procedure to the experimental data. The dynamics of the energy shifts were

analyzed with exponential functions. **d** Cartoon of electrostatic model estimating the transient valence band and HOMO shifts. Interfacial charge transfer creates a hole located on $C_{60}$ and an electron located initially on the first layer of WSe₂, which we assume to form physical dipoles across the interface (solid black equipotential lines). Neighboring dipoles interact by depolarization (dashed black arrows), reducing each other's magnitude in close proximity (near field region, NF), while creating an increased electrostatic potential far above the dipole layer (far field, FF). Since the origin of the dipole is located between $C_{60}$ and WSe₂, the Stark shifts experienced in the near field region have opposite signs for the two sides of the interface.

Initially, upon optical excitation ($t = 0$ fs), no changes in the energy level alignment and the spin polarization are observed for any of the WSe₂ valence bands, and all spectral changes can be attributed to an instantaneous inhomogeneous linewidth broadening caused mainly by the formation of the $C_{60}$-based CT₂ exciton. However, once interlayer charge transfer takes place and the charge-separated state is created ($t = 950$ fs), the WSe₂ valence bands shift. In both spin channels, the valence bands of the first layer (green Gaussian curves) transiently shift rigidly towards larger binding energies by $(50 \pm 20)$ meV, while the valence bands of the second layer reveal only a minor shift of $(20 \pm 20)$ meV. Thus, interlayer charge transfer at the $C_{60}$/WSe₂ heterointerface modifies the band structure in a layer-dependent fashion creating a transient spin polarization in the otherwise spin-unpolarized WSe₂. Crucially, the electric field gradient within the first two WSe₂ layers is strong enough to lead to a sizeable relative shift of the spin-polarized bands of the first vs. the second WSe₂ layer, thus creating a transient ferromagnetic-like spin polarization in the WSe₂ valence bands by revealing the hidden spin polarization in the surface region of the bulk crystal on ultrafast timescales.

Our experimental findings even allow us to estimate the magnitude of the relative shift of the valence band of the first and second WSe₂ layers for different interfacial charge carrier densities. This can be done by considering (i) the established linear relationship between

the amount of charging of the $C_{60}$ layer and the valence band shift of the first WSe₂ layer, and (ii) the layer-dependent valence band shift of WSe₂ for a selected charge density in the $C_{60}$ layer (see Supplementary Fig. S9 for more details). We find a linear increase in the relative spin splitting of the first and second layer WSe₂ valence band with increasing hole density in the $C_{60}$ layer, which can be as large as 60 meV for optical charge doping densities of about $6 \times 10^{13}$ cm⁻² ($6 \times 10^{-3}$ Å⁻²) in the $C_{60}$ layer. This spin-splitting and the associated optically-generated spin polarization is on the same order of magnitude as that observed for alkali metal doped WSe₂ bulk crystals with comparable dopant concentrations[40], and WSe₂ bilayers exposed to an external out-of-plane electric field[19]. Hence, our optical band structure engineering approach provides a clear pathway for creating sizeable transient spin polarizations in centrosymmetric systems that can be exploited for the realization of ultrafast spin functionalities.

In conclusion, our work has demonstrated a novel approach to transiently engineer the spin-polarized valence band structure in the otherwise spin-degenerate layered bulk material 2H-WSe₂. Specifically, the ultrafast electron transfer from an optically excited $C_{60}$ layer grown on top of WSe₂ leads to a layer-dependent shift of the spin-valley-layer locked WSe₂ valence band structure that ultimately reveals the hidden spin polarization of the system on a femtosecond timescale. Our optical manipulation scheme for generating a

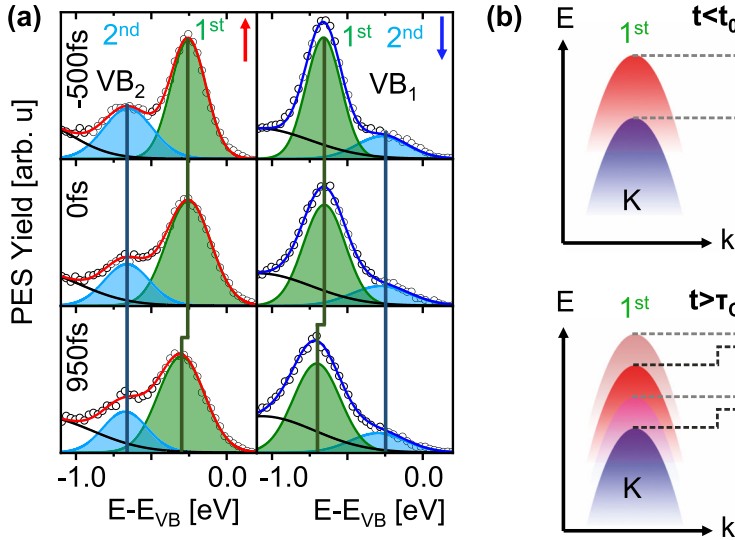

**Fig. 4 | Ultrafast changes of the hidden spin polarization of the WSe₂ bulk band structure. a** Time- and spin-resolved photoemission yield (out-of-plane spin component) of the valence band structure (see white dashed line) at three characteristic time delays. The data were recorded at the same electron momentum as the static data in Fig. 1c. The red and blue curves represent the fit to the spin-up and spin-down spectrum, respectively. The contributions of the first and second-layer valence bands to the spectral yield are fitted and illustrated as green and blue Gaussian curves underneath the spectra. The vertical solid lines indicate the significantly larger shift of the valence band of the first WSe₂ layer compared to the second layer. **b** Energy level diagram illustrating the ultrafast changes of the layer- and spin-dependent WSe₂ valence band structure after optical excitation with 3.2 eV photons. The gray horizontal dashed lines indicate the energy level alignment in the ground state, i.e., before the optical excitation ($t < t_0$), while the black dashed lines highlight the layer-dependent shift of the first and second layer WSe₂ valence band after optical excitation ($t > t_{CT}$).

ferromagnetic-like spin polarization in the valence band without an external magnetic field constitutes a new avenue to optically create ultrafast spin polarization in materials, in contrast to the typically observed quenching of charge and spin order following ultrafast laser excitation. This enables the optical engineering of new spin functionalities, such as the generation of spin-polarized hole currents from unpolarized DC charge currents in WSe₂, on ultrafast, sub-picosecond timescales, but also opens the intriguing possibility for exploiting and manipulating the orbital degree of freedom of layered TMDs thus paving the way for pushing the emergent field of orbitronics[41] towards ultrafast timescales.

## Methods

### Sample preparation

All sample preparation and measurement steps were performed under ultrahigh vacuum (UHV) conditions. The WSe₂ single crystals were obtained from HQ graphene and cleaved prior to the experiments resulting in a clean and flat surface. C₆₀ molecules were evaporated onto the surface at a pressure <10⁻⁸ mbar using a Knudsen-type evaporation source (Kentax GmbH). The molecular flux was calibrated using a quartz crystal oscillator gauge and the molecular coverage was estimated using the integrated intensity signal of the HOMO of C₆₀ as a reference.

### Spin- and time-resolved angle-resolved photoemission spectroscopy (ARPES)

The multidimensional photoemission experiments were conducted with a hemispherical analyzer (SPECS Phoibos 150) that is equipped with both a CCD detector system and the commercial spin detector (Focus FERRUM[42]) that is mounted in a 90° geometry after the hemispherical analyzer's exit slit plane. All spin-resolved photoemission data were recorded for the out-of-plane spin component, i.e., the spin component parallel to the optical axis of the analyzer lens optics. The spin sensitivity or Sherman function (S) of this very-low-energy electron diffraction detector was determined to be 0.29 for the out-of-plane spin component.

As excitation sources, we used the monochromatic He Iα radiation (21.2 eV, Scienta VUV5k) of a high-flux He discharge source as well as a pulsed femtosecond extreme ultraviolet (fs-XUV) light source. The fs-XUV radiation (22.2 eV, horizontal (p) polarization) was obtained by high harmonic generation (HHG) using the second harmonic (390 nm) of a titanium sapphire laser amplifier system (repetition rate 10 kHz, pulse duration <40 fs) to drive the HHG process[43]. The optical excitation of the organic material was also performed with the second harmonic of the amplifier system ($3.17 \pm 0.04$ eV, bandwidth 80 meV, horizontal ($p$) polarization). The beam size (diameter) of the pump and probe beam on the sample surface was determined to be $(500 \pm 10)$ μm in almost normal incidence geometry. Prior to each time-resolved experiment, the spatial overlap between the pump and the probe pulse was optimized directly on the sample plate, which was placed at the focus position of the analyzer. The spatial overlap was actively stabilized during the experiment to correct for spatial drift of the pump and probe beams. This is achieved by constantly monitoring the beam position of the fundamental laser beam at two well-defined positions in the laser beamline using two CCD cameras. Any lateral draft of the laser beam is compensated by two motorized mirrors installed in the beamline. All time-resolved photoemission experiments were conducted in a close-to-normal incidence geometry and an emission angle of ~35°. A detailed description of the data analysis procedure can be found in the Supplementary Information, Figs. S3–S7.

## Data availability

Relevant data supporting the key findings of this study are available within the article and the Supplementary Information file. All raw data generated during the current study are available from the corresponding authors upon request.

## Code availability

The electrostatic model calculations were performed with a dedicated code, which is available from the corresponding authors on request.

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

## Acknowledgements

The experimental work was funded by the Deutsche Forschungsgemeinschaft (DFG, German Research Foundation)—TRR 173—268565370 Spin+X: spin in its collective environment (Project A02) and by the Air Force Office of Scientific Research under award FA9550-21-1-0219. B. Stadtmüller acknowledges financial support from the Dynamics and Topology Center funded by the State of Rhineland Palatinate.

## Author contributions

The experiments were planned and supervised by B.S., O.L.A.M, and M.A. The time-and spin-resolved ARPES experiments were performed by B.A., S.L.Z., and S.H. The data were analyzed by B.A., S.H., and B.S. The electrostatic model simulations were carried out by S.L.Z and O.L.A.M. The results of the data analysis and simulations were discussed by all authors. The manuscript was written by B.A., O.L.A.M., and B.S. and discussed with all authors.

## Funding

## Competing interests

The authors declare no competing interests.
