## [Peer Review File · Nature Communications]

Revealing Hidden Spin Polarization in Centrosymmetric van der Waals Materials on Ultrafast TimescalesREVIEWER COMMENTS

Reviewer #1 (Remarks to the Author):

In this manuscript, the authors cleverly discerned spin- and layer-dependent valence band structure of 2H-stacked WSe₂ by spin- and time-resolved angle-resolved photoemission spectroscopy (ARPES). And by building layer-dependent interfacial electric field by ultrafast charge transfer, they broken the energy degeneracy of interfacial two adjacent WSe₂ layers and revealed the hidden spin polarization in the surface. This article is interesting, but flawed. I recommend it for publication after the following several questions are addressed:

- 1) I appreciate the application of spin- and time-resolved angle-resolved photoemission spectroscopy (ARPES) in bulk-WSe₂/ml-C60 interface. Due to the extreme surface sensitivity of the photoemission process, the results are dominated by interfacial properties not the bulk WSe₂. This avoids the difficulty in preparing large-area 2L-WSe₂/C60. However, the authors supposed all the signals of WSe₂ are attributed to interfacial two adjacent WSe₂ layers and the signal of top WSe₂ layer is about 3 times as much as the one of second WSe₂ layer. The authors should explain it in details for wider readers.
- 2) The size of light spot is missing.
- 3) The fluence is so high fluence that 40% C60 was excited according to Fig.2C. The extreme high fluence of 3.2 eV pump light (0.5 mJ/cm²) would exacerbate scattering and recombination process and bring lots of heat, which might influence the dynamics of excited carriers and electronic structure. The results might deviate from intrinsic properties. It is better to reduce the fluence by an order of magnitude.
- 4) The analysis of the dynamics of is based on two assumptions: the excitation light hardly excited WSe₂ layer and the electron in CT2 of C60 did not transfer to Σ -valley. In my opinion, those are illogical and influence both the analysis and the discussion of the data. The Fig.S2 shows the photoexcited electrons in conduction band of WSe₂ and the electrons in Σ -valley appear at the same time with the ones in K-valley in Fig.2c, which are contrary to the author's assumptions. According to those, the timescales in Figure.2C are incorrect.
- 5) Line 200 "Figure4.a" instead of "Figure.3a"; Line 207, "figure 4b" instead of "Figure.3b"
- 6) According to Fig.3c the lifetime of transient energy shift of WSe₂ VB is short as 700 fs. Why it is so short? Whether it is caused by the ultrafast relaxation process at this extremely high fluence?
- 7) The authors should establish the relationship between the densities of interfacial charge separated excitons and the relative shift of the spin-polarized bands of the first vs the second WSe₂ layer.
- 8) How about the kinetics and lifetime of the relative shift of the spin-polarized bands of the first vs the second WSe₂ layer? The authors should try to extract those data due to the importance for applications.
- 9) Fig.3d is hard to understand and Fig.4d needs to be optimized.

Reviewer #2 (Remarks to the Author):

The manuscript written by Arnoldi et al. investigated the transient spin polarization on a femtosecond timescale of the centrosymmetric 2H-stacked WSe₂. They have triggered an ultrafast interlayer electron transfer from the fullerene layer into the WSe₂ crystal, which subsequently creates an interfacial electric fields that holds the key to establishing layer-dependent transient spin

polarization. This work realizes the transient engineering of the spin-polarized valence band structure without an external magnetic field. The approach is interesting and the conclusion is clear. I would consider to recommend it for publication if authors can well address the following questions.

1. Hidden spin polarization emerges when the bulk space group of a solid has inversion symmetry (being centrosymmetric). However, authors constitute the C60/WSe₂ heterostructure, which will arise the structural inversion asymmetry and lead to bulk Rashba effect. How to clarify the influence of the Rashba effect on the spin polarization? How about the spin texture of the heterostructure? Please give more discussion on this issue.

2. Authors highlighted the key role of generating transient spin polarization in the Abstract and Introduction parts. In this manuscript, the discussion of transient band structure engineering should be strengthened in the main text, such as the comparison with other approach or significance of this result.

3. The Hamiltonian of the 2H-stacked WSe₂ is given by equation (1) according to Ref. [17]. I'm not sure this equation is suitable for the C60/WSe₂ heterostructure as the interlayer electron transfer should be considered. Can authors relate this equation to the experimental results?

4. Is there any spin-polarized current or charge current generated in C60/WSe₂ heterostructure?

5. What is the motivation of selecting fullerene to create the interface? Are there any selection criteria?

6. Authors stated "Optical excitation of the heterostructure with 3.2 eV sub-50 fs pulses", but the photon energy in Fig. 2(a) is 3.1 eV. Why is that?

7. More details of Fig.3 (d) should be given in the figure caption for readers to understand the valence band shift.

Point-to-point reply to the reviewers' comments.

Reviewer #1

In this manuscript, the authors cleverly discerned spin- and layer-dependent valence band structure of 2H-stacked WSe₂ by spin- and time-resolved angle-resolved photoemission spectroscopy (ARPES). And by building layer-dependent interfacial electric field by ultrafast charge transfer, they broken the energy degeneracy of interfacial two adjacent WSe₂ layers and revealed the hidden spin polarization in the surface. This article is interesting, but flawed. I recommend it for publication after the following several questions are addressed:

Comment 1.1

1) I appreciate the application of spin- and time-resolved angle-resolved photoemission spectroscopy (ARPES) in bulk-WSe₂/ml-C60 interface. Due to the extreme surface sensitivity of the photoemission process, the results are dominated by interfacial properties not the bulk WSe₂. This avoids the difficulty in preparing large-area 2L-WSe₂/C60. However, the authors supposed all the signals of WSe₂ are attributed to interfacial two adjacent WSe₂ layers and the signal of top WSe₂ layer is about 3 times as much as the one of second WSe₂ layer. The authors should explain it in details for wider readers.

Answer to Comment 1.1:

The high surface sensitivity of the photoemission experiment is due to the small elastic mean free path of electrons at low kinetic energies, i.e. $E_{kin} < 50$ eV. In our study, we take advantage of the fact that the elastic mean-free path of electrons in WSe₂ has been quantified in several independent experiments. For example, Riley et al. [Ref. 14 in the revised manuscript] used spin-resolved photoemission to uncover an elastic mean-free path of electrons of approximately one WSe₂ trilayer, i.e., 1 nm for photon energies in the 20-40 eV range. These results were confirmed by Parashar et al. [Ref. 26 in the revised manuscript], who directly compared the photoemission intensity of the valence band of the first and second tri-layers in a twisted bilayer of WSe₂. In the simplest model, the layer-dependent photoemission yield of a multilayer structure follows the Lambert-Beer law and decreases exponentially for buried layers. Using the elastic mean free path of electrons of 1 nm as a decay constant, we find that the photoemission yield of the second WSe₂ layer is about 1/3 of the yield of the first layer. This estimate is fully consistent with the relative ratio of photoemission intensity of 3:1 of our spin-resolved photoemission data in Fig. 1 in the main text or Fig. S2 in the Supplementary Information.

In the revised version of the manuscript, we explicitly mention the exponential attenuation of the photoemission intensity for buried layers and the extremely small elastic mean free path of electrons in WSe₂. We also cite the work of Riley et al. and Parashar et al. in this context.

Changes in the manuscript in answer to Comment 1.1:

- We mention the exponential attenuation of the photoemission intensity for buried layers in the main manuscript, and in the caption of Fig. 1 in the main text and Fig. S2 in the supplementary information, see page 7, lines 129 - 131 of the main manuscript, caption of Fig 1 on page 6 of the main manuscript, and caption of Fig S2 on page 3 of the supporting information.

Comment 1.2

2) The size of light spot is missing.

Answer to Comment 1.2:

The spot size diameter of the pump beam on the sample surface is approximately $d \approx 500 \mu\text{m}$, and that of the probe spot is also $d \approx 500 \mu\text{m}$. These values were determined for the experimental geometry used to obtain the photoemission signal around the K-point of WSe_2 , i.e. in almost normal incidence geometry. We have added this information to the Methods section of our manuscript. Note that we carefully checked the spatial and temporal overlap of the beam and probe beam prior to each experiment, as mentioned in the Method section of our manuscript (see lines 303/304).

Changes in the manuscript in answer to Comment 1.2:

- We mention the beam spot diameter of the pump and probe beam on the sample surface in the *Methods* section, see lines 303 – 304 on page 18 of the main manuscript.

Comment 1.3

3) The fluence is so high fluence that 40% C60 was excited according to Fig.2C. The extreme high fluence of 3.2 eV pump light (0.5 mJ/cm^2) would exacerbate scattering and recombination process and bring lots of heat, which might influence the dynamics of excited carriers and electronic structure. The results might deviate from intrinsic properties. It is better to reduce the fluence by an order of magnitude.

Answer to Comment 1.3:

The reviewer is correct that the intrinsic charge carrier dynamics of single layer and bulk TMDCs are typically studied in the so-called “low fluence regime” using fluences in the range of tens of $\mu\text{J/cm}^2$. This regime allows a clear view, for instance, of the exciton dynamics and their (momentum-dependent) dephasing time after optical excitation. These properties have already been studied extensively for WSe_2 and other TMDCs before.

However, these fluences are much too small to create a sufficient charge transfer across the interface to alter the transient energy level alignment at the $\text{C}_{60}/\text{WSe}_2$ interface. As a consequence, we had to rely on applied fluences of up to $500 \mu\text{J/cm}^2$ to excite and transfer enough charges across the interface to observe significant transient energy shifts of the valence bands of both sides of the interface. This is also discussed in our reply to review comment 1.7 where we correlate the number of charges at the interface to the magnitude of the transient energy shift of the WSe_2 valence bands. Therefore, repeating the experiment with an order of magnitude smaller fluence would lead to a complete disappearance of the transient shifts at the interface and hence would not contribute to refining our understanding of the phenomenon discussed in our manuscript.

Finally, we would like to point out that large fluences such as ours lead only to small deviations in the intrinsic carrier dynamics of WSe_2 , as already reported by M. Puppini et al [see Ref 3 in the supporting information, <http://dx.doi.org/10.17169/refubium-804>]. The fluence in our study is still well below the critical Mott density where excitons are suppressed. This is confirmed by the charge carrier dynamics of the bare WSe_2 reported in the supporting information.

In the revised version of the manuscript, we now explicitly mention that the large fluences of the optical excitation are necessary to realize the transient band structure changes at the interface. In addition, we

mention in the supporting information that the fluences are still smaller than the crucial Mott density of WSe_2 .

Changes in the manuscript in answer to Comment 1.3:

- We mention that the fluence values in the manuscript and SI refer to the applied fluence, **see line 139 and the caption of Fig. 2 on pages 7 and 6 of the revised manuscript, as well as captions of Figs. S3 and S4 on pages 4 and 6 of the supporting information.**

- We explicitly state that the hot carrier dynamics in the conduction band structure of WSe_2 observed for large applied fluences is qualitatively similar to the one reported in the literature for small applied fluences, **see caption of Fig. S3 on page 4 in the supporting information.**

- We added a statement that the large applied fluence of our experiment is still smaller than the critical Mott density below which excitons are suppressed in WSe_2 , **see caption of Fig. S3 on page 4 in the supporting information.**

Comment 1.4

4) The analysis of the dynamics of is based on two assumptions: the excitation light hardly excited WSe_2 layer and the electron in CT_2 of C_{60} did not transfer to Σ -valley. In my opinion, those are illogical and influence both the analysis and the discussion of the data. The Fig.S2 shows the photoexcited electrons in conduction band of WSe_2 and the electrons in Σ -valley appear at the same time with the ones in K-valley in Fig.2c, which are contrary to the author's assumptions. According to those, the timescales in Figure.2C are incorrect.

Answer to Comment 1.4:

The reviewer raises an important point that was too briefly addressed in our original manuscript. We indeed employed certain simplifications in our rate equation model that did not allow us to capture the complete excited state population dynamics of the $\text{C}_{60}/\text{WSe}_2$ interface. However, these simplifications do not affect the interpretation of our data. The reason for this is that our model was mainly designed to capture the most significant differences between the hot electron dynamics of the bare and C_{60} -covered surfaces, namely the substantial transient population of the conduction band valley at the K-point that only occurs for the $\text{C}_{60}/\text{WSe}_2$ interface. The dynamics of these carriers are the ones responsible for the transient valence band shifts at the $\text{C}_{60}/\text{WSe}_2$ interface, as discussed in the manuscript and our detailed answers to the reviewer's questions 1.6 and 1.7. In general, the dynamics of the carriers at the K-point can also be quantified by a simple exponential model of the time-dependent photoemission yield at the K-point. The results of such a fit are fully consistent with those from our rate equation model at the K-point. We find a population time of (100 ± 50) fs and a depopulation time of (1350 ± 50) fs. This confirms that our rate equation model does not depend on the specifics of the complete interfacial dynamics and that it captures the time evolution of the electron population at the K-point.

In contrast to the dynamics at the K-point, we implemented certain simplifications when describing the population dynamics in the Σ -point of the conduction band. As pointed out by the reviewer, we neglected the interlayer charge transfer between the CT_2 state of C_{60} and the Σ -point of WSe_2 , as well as the intrinsic population dynamics of WSe_2 that is triggered by 3.2 eV photons. This was done to minimize the number of fitting parameters in the model.

In the revision process, we have now extended our rate equation model to include both the intrinsic excitation of a population of carriers in the WSe₂ conduction as well as the possible interlayer transfer from C₆₀ into the Σ -point of WSe₂.

As illustrated in Fig. S2 of the supplementary information, the optical excitation of WSe₂ with 3.2 eV photons also leads to the appearance of carriers in the Σ -point of the conduction band due to rapid scattering from the Γ -point region within the first 1ps. This has also been previously reported by Puppini et al. [27 in the revised manuscript]. In order to include these intrinsic WSe₂ dynamics, we now add one additional scattering channel in our rate equation model that models the intraband scattering of optically excited carriers from the WSe₂ conduction band into the conduction band valley at the Σ -point (see Fig. S8). The best fitting results were obtained for a scattering time of approx. 1.1ps. This value is comparable to the intraband scattering time obtained for bare WSe₂ after optical excitation with blue light as determined by our reference experiment and reported by Puppini et al. [Ref. 27 in the revised manuscript, and Ref. 3 and 4 in the supporting information].

Moreover, we also add another direct scattering path from the optically excited carriers in C₆₀ into the WSe₂ Σ -point of the conduction band, as suggested by the reviewer. The best fitting result was obtained for the case that 10% of carriers scatter into the Σ -point while the majority of carriers (90%) still scatter into the K point.

Both extensions of our rate equation model improve the description of the increase in population observed at the conduction band valley at the Σ -point, see Fig. 2c, but do not fundamentally alter our conclusions, nor the interpretation of the observed transient band shifts.

In the revised version of the manuscript, we now present the extended version of our rate equation model. We have also improved the discussion of the role of the intrinsic charge carrier dynamics of WSe₂ for the transient valence band shifts. However, we point out again that *these changes in the rate equation model do not in any way impact the central findings and interpretations of the initial version of our manuscript.*

Changes in the manuscript in answer to Comment 1.4:

- We have adapted our rate equation model and its description in the supporting information, **see Supplementary Methods A on pages 14 and 15 of the supporting information.**
- We have updated the graphical illustration of the scattering rates of our rate equation model and have replaced the fitted population curves of our rate equation model in Fig. 2 and Fig. S8, **see Fig. 2 on page 8 of the main text and Fig. S8 on page 11 of the supporting information.**
- We have improved the discussion of the role of the intrinsic electron dynamics in WSe₂ for the interfacial population dynamics and for the formation of the charge-separated state the C₆₀/WSe₂ interface, **see lines 170-178 on page 11 and lines 196 – 199 on page 12 of our main manuscript.**

Comment 1.5

5) Line 200 “Figure4.a” instead of “Figure.3a”; Line 207, “figure 4b” instead of “Figure.3b”

Answer to Comment 1.5:

We thank the reviewer for her/his observation. We have changed this typo.

Changes in the manuscript in answer to Comment 1.5:

- We have replaced Fig. 3a/b with Fig. 4a/b, see lines 227 and 234 on pages 14 and 15 of the main manuscript.

Comment 1.6:

6) According to Fig.3c the lifetime of transient energy shift of WSe₂ VB is short as 700 fs. Why it is so short? Whether it is caused by the ultrafast relaxation process at this extremely high fluence?

Answer to Comment 1.6:

We thank the reviewer for raising this point, which was not discussed explicitly enough in our original version of the manuscript. As pointed out on page 13, the transient valence band shifts can be understood by our simple electrostatic model simulation, which was mainly discussed in the Supplementary Information of our manuscript. In essence, our model shows that any charging of the interface layers, i.e., the C₆₀ and WSe₂ layers, leads to an electrostatic field that causes a Stark-like energy shift of the valence bands of the adjacent layer. The magnitude of this shift is directly related to the strength of the electric field and thus to the amount of charge in the adjacent layer.

According to our model, we can attribute the transient shifts of the WSe₂ valence bands to the transient charging of the C₆₀ layer caused by the excitation of the C₆₀ layer and the subsequent electron transfer into WSe₂. This theoretically proposed direct link between the valence band shifts and the transient charging of the advanced layer is also fully consistent with the carrier dynamics observed in our experiment. As shown in Fig. 2c of our main manuscript, the majority of holes in the C₆₀ layer disappear within the first (730±50) fs (fast recovery time constant), thereby significantly reducing the charging of the C₆₀ layer. This depopulation time is in perfect agreement (within experimental uncertainty) with the recovery of the transient energy shift of the WSe₂ valence band. Finally, we note that this reasoning also holds for the recovery time of the C₆₀ valence band shift of (1380 ±50) fs. This timescale is in excellent agreement with the intervalley scattering time from the K to the Σ point of the WSe₂ conduction band (~1350 fs), which is responsible for the delocalization of carriers within WSe₂.

In the main text of the revised manuscript, we now provide more details on the relationship between the transient charging of the interfacial layer and the corresponding valence band shifts of the adjacent layer.

Finally, we would like to point out that the charge carrier population dynamics are in no way affected by the fluence used in our experiment. This is explained in more detail in our response to reviewer comment 1.3.

Changes in the manuscript in answer to Comment 1.6:

- We have revised the paragraph of our main manuscript discussing the transient valence band shifts of WSe₂ and the C₆₀ layer, see lines 209 – 216 on page 13 of the main manuscript.

Comment 1.7:

7) The authors should establish the relationship between the densities of interfacial charge separated excitons and the relative shift of the spin-polarized bands of the first vs the second WSe₂ layer.

Answer to Comment 1.7:

The reviewer raises an interesting question, which cannot be answered by our spin- and time-resolved photoemission experiment alone, as pointed out below. Instead, we combine our time-resolved but **spin-integrated** photoemission data with our time- and spin-resolved photoemission data to provide an estimate of the relative shift between the spin-polarized WSe_2 and the charge carrier density at the interfaces. As pointed out in our **answer to comment 1.6**, the magnitude of the shift of the WSe_2 valence band is directly correlated to the number of charges in the C_{60} layer. This allows us to correlate the (transient) energy shift of the WSe_2 valence band and the (transient) hole density within the C_{60} layer, see Fig. R1. The transient energy shift in the spin-integrated photoemission data (see also Fig. 3c) mainly reflects the behavior of the valence band of the first WSe_2 layer. This is due to the small elastic mean free path of photoelectrons at low kinetic energies, see our answer to comment 1.1. The density of holes in the C_{60} layer was determined from the transient loss of intensity of the HOMO signal (see Fig. 2c), the coverage of C_{60} of 0.8ML, a maximal packing density of C_{60} molecules on surfaces of one C_{60}

Fig. R1. Energy shift of the WSe_2 valence band vs. the hole density within the C_{60} layer. These values were extracted from the time-resolved, but spin-integrated photoemission data set shown in Figs. 2c and 3c in our manuscript. The density of carriers (holes) in the C_{60} layer was determined from the transient loss of intensity of the HOMO signal shown in Fig. 2c of the main manuscript, the coverage of C_{60} of 0.8 ML, a maximum packing density of C_{60} molecules on surfaces of one C_{60} molecule per 100 \AA^2 , and an excitation density of 70% of all C_{60} molecules. We find a linear relationship between the hole density and the valence band energy shift of the first layer (red line), with a turn-off below $0.001 \text{ charges/\AA}^2$. This linear relationship between charge density and valence band shift of the first WSe_2 layer, together with our spin- and time-resolved photoemission data (see Fig. 4 of the main manuscript), allows us to estimate the magnitude of the relative valence band shift of the first and second WSe_2 layers. For a given hole density in the C_{60} layer, we find a relative shift of the valence band of $(50 \pm 20) \text{ meV}$ for the first (red dotted lines) and $(20 \pm 20) \text{ meV}$ for the second WSe_2 layer (blue dotted lines) in our spin- and time-resolved photoemission experiment. Assuming a similar linear relationship between hole density and valence band shift for the second WSe_2 layer, as well as a similar turn-off hole density, we can estimate the transient valence band shift of the second WSe_2 layer, see blue solid line in Fig.R1.

molecule per 100\AA^2 , and an excitation density of 70% of all C_{60} molecules. We find a linear relationship between the hole density and the energy shift of the valence band of the first layer (see red line in Fig. R1), with a turn-off below $0.001\text{ charges/\AA}^2$. To estimate the relative shift of the first and second WSe_2 layers, we consider the findings of our time- and spin-resolved photoemission experiment that are presented in Fig. 4 of our main manuscript. For a specific hole density in the C_{60} layer, we find a relative shift of the valence band of $50\pm 20\text{meV}$ for the first and $20\pm 20\text{meV}$ for the second WSe_2 layer. To then estimate the relative shift of the valence band of 2nd WSe_2 layer, we assume *i)* that the relationship between the hole density and the magnitude of the valence band shift of the 2nd WSe_2 layer is also linear, and *ii)* that the shift of the WSe_2 valence band of the 2nd layer starts at the same minimal hole density as in the first layer (i.e. the turn-off around $0.001\text{ charges/\AA}^2$). With only these two assumptions, and the experimental measurement of the valence band shifts of first and second WSe_2 layer, we can estimate the relationship between the hole density in the C_{60} layer and the valence band shift of the second WSe_2 layer. The corresponding slope is included in Fig. R1 as a blue solid line.

In order to make this clearer in the manuscript, we newly present and discuss Fig. R1 and the above rationale as Fig. S9 in the supporting information of our manuscript.

To justify our approach, we would like to point out to the reviewer that a reliable experimental determination of the relative shift of the valence band of the first and second WSe_2 layers for different hole densities in C_{60} requires an unfeasibly extensive data set of spin- and time-resolved photoemission spectra at the K-point of the C_{60}/WSe_2 heterostructure for different excitation strengths and/or different time delays. This is extremely challenging and time-consuming, even with the state-of-the-art spin-resolved photoemission technology used in our experiment, making such experiments nearly impossible. To emphasize this fact, we would like to point out that each single spin-resolved photoemission spectrum (spin-up or spin-down) at one time delay (see Fig. 4a) required a data acquisition time of nearly 10-12h (including realignment of the laser beamline and the experiment). This extremely long data acquisition time is necessary to achieve the required signal-to-noise ratio for a reliable line shape analysis, i.e. for a reliable fitting of the experimental data with Gaussian functions. In addition, our experiments were performed at low sample temperatures using liquid helium cooling to increase the depopulation time of the optically excited carriers (electrons and holes) at the C_{60}/WSe_2 interface. This was necessary to increase the recovery time of the transient valence band shifts to a few hundred femtoseconds and to make our experimental results more convincing to the reader of our manuscript. Unfortunately, the limited size of our liquid helium reservoir limits the total data acquisition time to a few days. This allowed us to record only the spin-resolved photoemission data shown in the manuscript.

We hope therefore that the reviewer appreciates the extreme experimental challenges that make it rather impossible to obtain substantially larger spin- and time-resolved photoemission data sets. However, we are convinced that our estimation of the relative valence band shift of the first and second WSe_2 layer for different hole densities in C_{60} based on our spin-integrated but time-resolved photoemission data in conjunction with our already existing time- and spin-resolved photoemission data provides a similarly deep understanding of the ultrafast band structure dynamics of the first WSe_2 layers.

Changes in the manuscript in answer to Comment 1.7:

- We have added Fig. R1 as Fig. S9 to the supporting information and discuss the relationship between

the layer-dependent valence band shift in WSe₂ and the charge carrier density in the C₆₀ layer in the corresponding caption, **see Fig. S9 and caption on page 12 of the supporting information.**

- We discuss the linear relationship between the charge density in the C₆₀ layer and the valence band shift of the first WSe₂ layer in the revised paragraph of our main manuscript discussing the transient valence band shifts, **see lines 212 – 214 on pages 13 of the main manuscript.**

- We discuss the linear increase in the spin splitting of the first and second layer WSe₂ valence band structure with increasing charging of the C₆₀ layer and provide an estimate of the relative shift of the WSe₂ valence bands for the maximum carrier density used in our experimental study, **see lines 249 – 262 on pages 15 and 16 of the main manuscript.**

Comment 1.8:

8) How about the kinetics and lifetime of the relative shift of the spin-polarized bands of the first vs the second WSe₂ layer? The authors should try to extract those data due to the importance for applications.

Answer to Comment 1.8:

Our manuscript, i.e., our experimental results and our theoretical model, provide clear evidence that the transient band structure changes of both sides of the interface are determined by the interfacial carrier dynamics. This is also emphasized in our **answers to the reviewer's questions 6 and 7**. In particular, we now show a linear relationship between the number of holes in the C₆₀ layer and the shift of the WSe₂ valence band feature in the spin-integrated photoemission study, which is dominated by the contributions of the first WSe₂ layer. Therefore, our experiment directly shows that the lifetime and dynamics of the valence band shift of the first WSe₂ layer follow the population dynamics of the holes in C₆₀. This allows us to estimate the relative shift of the spin-polarized bands of the first and second layers; see answers to reviewer questions 7. However, the "lifetimes" and dynamics of the valence band shifts of the first and second WSe₂ layers are identical, since both shifts are also related to the same population dynamics of the holes in C₆₀. All these aspects have been addressed in our reply to the reviewer's questions 6 and 7.

Changes in the manuscript in answer to Comment 1.8:

- see changes in the manuscript in answer to the reviewer's comments 1.6 and 1.7.

Comment 1.9

Fig.3d is hard to understand and Fig.4d needs to be optimized.

Answer to Comment 1.9:

We have improved the presentation of Fig. 3d. In addition, we have extended the description of this figure in the caption of Fig.3. Unfortunately, it is unclear to us which other figure should be optimized as our manuscript does not contain a Fig. 4d.

Changes in the manuscript in answer to Comment 1.9:

- We have optimized Fig. 3d of our initial manuscript and have improved the description of the Figure in the caption of Fig. 3, **see page 10, Fig.3 and caption of the main manuscript.**

Fig. R2. Improved version of Fig. 3d.

Reviewer #2

The manuscript written by Arnoldi et al. investigated the transient spin polarization on a femtosecond timescale of the centrosymmetric 2H-stacked WSe₂. They have triggered an ultrafast interlayer electron transfer from the fullerene layer into the WSe₂ crystal, which subsequently creates an interfacial electric fields that holds the key to establishing layer-dependent transient spin polarization. This work realizes the transient engineering of the spin-polarized valence band structure without an external magnetic field. The approach is interesting and the conclusion is clear. I would consider to recommend it for publication if authors can well address the following questions.

Comment 2.1:

1. Hidden spin polarization emerges when the bulk space group of a solid has inversion symmetry (being centrosymmetric). However, authors constitute the C₆₀/WSe₂ heterostructure, which will arise the structural inversion asymmetry and lead to bulk Rashba effect. How to clarify the influence of the Rashba effect on the spin polarization? How about the spin texture of the heterostructure? Please give more discussion on this issue.

Answer to Comment 2.1:

We thank the reviewer for this interesting comment. The bulk and surface Rashba effects both arise from broken structural inversion symmetry in material systems with strong spin-orbit coupling. It typically leads to a momentum-dependent spin splitting of the otherwise spin-degenerate bands around the Γ -point of the band structure, i.e. to a relative shift of the otherwise spin-degenerate bands in momentum space. This spin splitting is typically accompanied by spin-momentum locking, where the orientation of the spin is perpendicular to the momentum of the electron.

In the case of the bare WSe₂ bulk crystal or the C₆₀/WSe₂ heterostructure, the structural inversion symmetry is broken at the surface of the WSe₂ bulk crystal, which could lead to Rashba-type spin splitting of the band structure in the surface region around the Γ -point. However, we observe no evidence for momentum splitting of the valence bands of WSe₂ and thus no evidence for the Rashba effect in this system, in agreement with the published results of the group of P. King [Riley et al, Ref. 14 in the revised manuscript]. This shows that the potential gradient at the surface or at the C₆₀/WSe₂ interface is not strong enough to cause significant Rashba-type spin splitting of the bands in the surface region of WSe₂.

Instead, we find a clear out-of-plane spin polarization of the WSe₂ valence bands near the K-point for both the bare and the C₆₀-covered surface. This is the clear signature of the hidden spin polarization, as previously demonstrated by Riley et al, [Ref. 14 in the revised manuscript]. Accordingly, our experimental results contain only clear signatures of the changes in the hidden spin polarization.

Finally, we would like to point out that even our optically induced band structure changes lead to a layer-dependent change of the valence band energy and we find no indication of a Rashba-type spin splitting in the WSe₂ valence band structure. To clarify this issue, we now specifically discuss the

potential role of the Rashba effect for the spin texture of the C_{60}/WSe_2 interface in the revised version of the manuscript.

Changes in the manuscript in answer to Comment 2.1:

- We specifically state that the valence band structure of WSe_2 is not altered by the adsorption of C_{60} and that no signature of a Rashba-type spin splitting was detected in the C_{60}/WSe_2 valence band structure, **see lines 114-118 page 7 of the main manuscript.**
- We have added ARPES data for bare and C_{60} -covered WSe_2 along the Γ -K direction to provide conclusive evidence that the valence band structure of WSe_2 remains unaffected by the adsorption of C_{60} , **see Fig. S1 on page 2 of the supporting information.**

Comment 2.2:

2. Authors highlighted the key role of generating transient spin polarization in the Abstract and Introduction parts. In this manuscript, the discussion of transient band structure engineering should be strengthened in the main text, such as the comparison with other approach or significance of this result.

Answer to Comment 2.2:

Indeed, our discussion of the transient spin polarization was too brief in the main part of the manuscript. We have improved our discussion of the transient generation of spin polarization in the paragraph discussing our time- and spin-resolved photoemission data. We now discuss the correlation between the (optically generated) density of interfacial charge carriers and the magnitude of the relative valence band shift of the valence band of the first and second WSe_2 layer, see also our answer to the reviewer comment 1.7. We further compare of the magnitude of the observed transient spin splitting in light of spin polarizations in layered materials that were created by charge doping (see Ref 40 of the revised manuscript) or external fields (see Ref 19 of the revised manuscript). We find that the magnitude of the spin splitting between the valence band structure of the first and second WSe_2 layer is on the same order of magnitude as reported for these static manipulation schemes. This underlines the significance of our findings and the potential to exploit the optically generated spin polarization for ultrafast spin functionalities.

Changes in the manuscript in answer to Comment 2.2:

- We provide a more detailed discussion of the optically-generated transient spin polarization in WSe_2 and compare our findings to static band structure engineering approaches in literature, **see lines 249 – 262 on pages 15 and 16 of the main manuscript.**
- We have added the publication, discussing the chemical band structure engineering of WSe_2 by alkali metal doping, to our list of references, **see Refs 40 in the reference list of our manuscript.**

Comment 2.3:

3. The Hamiltonian of the 2H-stacked WSe_2 is given by equation (1) according to Ref. [17]. I'm not sure this equation is suitable for the C_{60}/WSe_2 heterostructure as the interlayer electron transfer should be considered. Can authors relate this equation to the experimental results?

Answer to Comment 2.3:

In general, we agree with the reviewer that the Hamiltonian in equation (1) was derived for a bare 2H-

WSe₂ crystal and not for a C₆₀/WSe₂ heterostructure. Therefore, we agree that it is most likely unsuited for a quantitative description of the electronic properties and the spin-splitting of the valence band structure of the C₆₀/WSe₂ heterostructure. This is, however, not the intention of this equation: Rather, we use the Hamiltonian of the bare 2H-WSe₂ crystal as a conceptual framework for understanding coupling of the different degrees of freedom (spin, layer, valley) and as a vivid illustration of the different possibilities for manipulating the spin-dependent electronic properties of the 2H-WSe₂ crystal. We now mention this more clearly in the introduction of our manuscript.

Changes in the manuscript in answer to Comment 2.3:

- We mention that the Hamiltonian (equation. 1) does not necessarily describe the electronic properties of the C₆₀/WSe₂ heterostructure, but is used as a schematic illustration of the different possibilities to manipulate the electronic properties of the bare 2H-WSe₂ by external stimuli, **see lines 79 – 81 on page 4 of the main manuscript.**

Comment 2.4:

4. Is there any spin-polarized current or charge current generated in C₆₀/WSe₂ heterostructure?

Answer to Comment 2.4:

The optical excitation of the C₆₀/WSe₂ heterostructure leads to an ultrafast interfacial charge transfer from the C₆₀ layer into the WSe₂ bulk crystal. This charge transfer is unpolarized, but responsible for the transient spin-dependent band structure changes observed in our work. Strictly speaking, this interfacial charge transfer is not a charge current in the conventional sense of a current. In our conclusion, we propose that our band structure engineering approach can be exploited to generate a spin-polarized hole current in WSe₂ on ultrafast timescales. This, however, requires the existence of an unpolarized (DC-) hole current in the WSe₂ bulk crystal that could be polarized on ultrafast timescales by an optically induced change in the valence band structure, as demonstrated in our manuscript. Accordingly, we clarify our statement in the conclusion of our manuscript.

Changes in the manuscript in answer to Comment 2.4:

- We clarified our statement of the generation of an ultrafast hole current in the introduction and the conclusion of our manuscript, **see lines 98/99 and 272/273 on pages 5 and 16 of the main manuscript.**

Comment 2.5:

What is the motivation of selecting fullerene to create the interface? Are there any selection criteria?

Answer to Comment 2.5:

The fullerene C₆₀ was selected since it is one of the most commonly used molecular materials in optoelectronic and spintronics research. It is extremely robust against chemical degradation and irradiation with intense and ultrashort laser radiation. In addition, it is well established that the excited states of C₆₀ and other fullerenes are dominated by so-called charge transfer excitons where the optical excitation directly leads to a separation of the electron and hole of the excited exciton on neighboring molecular sites. These charge transfer excitons are expected to act as a precursor for charge separation and interfacial charge transfer as required for our study. This has already been mentioned briefly in the original version of our manuscript (see page 9). We now specifically mention this in the introduction of our manuscript.

Changes in the manuscript in answer to Comment 2.5:

- We state that C_{60} was selected due to the existence of charge transfer excitons after optical excitation that act as precursor for charge separation and charge transfer across interface, **see lines 94-96 on page 5 of the main manuscript.**

Comment 2.6:

6. Authors stated “Optical excitation of the heterostructure with 3.2 eV sub-50 fs pulses”, but the photon energy in Fig. 2(a) is 3.1 eV. Why is that?

Answer to Comment 2.6:

We thank the reviewer for her/his observation. The photon energy of the optical excitation is 3.2 eV. We have corrected our mistake in Fig. 2.

Changes in the manuscript in answer to Comment 2.6:

- We corrected the photon energy of the optical excitation in the energy level diagram. It is now 3.2 eV, **see Fig. 2 on page 8 of the main manuscript.**

Comment 2.7:

7. More details of Fig.3 (d) should be given in the figure caption for readers to understand the valence band shift.

Answer to Comment 2.7:

As pointed out in our answer to the reviewer comment 1.9, we have improved the presentation of Fig. 3d and have extended the description of the figure in the caption of Fig. 3.

Changes in the manuscript in answer to Comment 2.7:

- We have optimized Fig. 3d of our initial manuscript and have improved the description of the figure in the caption of Fig. 3, **see page 10, Fig.3 and caption of the main manuscript.**

Reviewers' Comments:

Reviewer #1:

Remarks to the Author:

Thank the authors for their detailed answers and corrections. The previous questions are well answered and this manuscript has been greatly improved. I recommend this work for publication.

There are some minors to be improved.

Some "see SI" in the main text are unclear. These need to be optimized to clearly refer to "Figure SX-SX" or "SI note X" for readers. For example, line 114, 117,154,165,167,175,178,203,207,214,253.

Reviewer #2:

Remarks to the Author:

The authors have made the revision. I have no other comments.

Point-to-point reply to the reviewers' comments. – Revision 2

Only reviewer #1 raised an additional minor issue in her/his second report. Therefore, we will only discuss her/his response in our response.

Reviewer #1

Reviewer #1 (Remarks to the Author):

Thank the authors for their detailed answers and corrections. The previous questions are well answered and this manuscript has been greatly improved. I recommend this work for publication. There are some minors to be improved.

Comment 1.1

Some “see SI” in the main text are unclear. These need to be optimized to clearly refer to “Figure SX-SX” or “SI note X” for readers. For example, line 114, 117,154,165,167,175,178,203,207,214,253.

Answer to Comment 1.1:

We thank the reviewer for appreciating our efforts to improve our manuscript. As suggested by the reviewer, we have clarified the ambiguous references in the main manuscript to the supporting information. This will allow the readers to follow our manuscript more easily.

Changes in the manuscript in answer to Comment 1.1:

- We have specified the link between the main manuscript and the supporting information in the following lines of our revised manuscript: Lines 112, 115, 153, 163/165, 166, 174, 177, 202, 206/207, 214, 255, 314.

Changes in Response to the Editorial Checklist

We have made the following changes to our manuscript in response to the editorial checklist provided by the editor. All changes are also highlighted in green color in the file “Arnoldi_etal_Revision2_Changes.pdf”.

1. We have modified the abstract according to the editor’s suggestion with only minor changes, see lines 17-32 of the main manuscript.
2. We have replaced the section title "Results and Discussion" with "Results" to comply with the editorial guidelines, see line 105. We have also repeated the captions of all four figures in the main manuscript at the end of the text, see lines 431 ff.
3. We have added explanations to the captions of Figs. 2,3, S7, and S9 to define the error bars.
4. We have improved our data availability statement, see lines 316 – 318.
5. We have updated the citation of the reference [23]. The preprint has been published in Nat. Commun.